# Fusarium Head Blight in Barley from Subtropical Southern Brazil: Associated *Fusarium* Species and Grain Contamination Levels of Deoxynivalenol and Nivalenol

**DOI:** 10.3390/plants14152327

**Published:** 2025-07-27

**Authors:** Emanueli Bizarro Furtado, Eduardo Guatimosim, Danielle Ribeiro de Barros, Carlos Augusto Mallmann, Jeronimo Vieira de Araujo Filho, Sabrina de Oliveira Martins, Dauri José Tessmann, Cesar Valmor Rombaldi, Luara Medianeira de Lima Schlösser, Adriana Favaretto, Leandro José Dallagnol

**Affiliations:** 1Departamento de Fitossanidade, Faculdade de Agronomia Eliseu Maciel, Universidade Federal de Pelotas, Capão do Leão 96160-000, RS, Brazil; emanuelifurtado@gmail.com (E.B.F.); danrbarros@hotmail.com (D.R.d.B.); jeronimo.vieira@ufpel.edu.br (J.V.d.A.F.); sabrina-martins11@hotmail.com (S.d.O.M.); 2Instituto de Ciências Biológicas, Universidade Federal de Rio Grande, São Lourenço do Sul 97300-000, RS, Brazil; e.guatimosim@furg.br; 3Departamento de Medicina Veterinária Preventiva, Centro de Ciências Rurais, Universidade Federal de Santa Maria, Santa Maria 97105-900, RS, Brazil; mallmann@lamic.ufsm.br (C.A.M.); lauraschlosser@hotmail.com (L.M.d.L.S.); 4Departamento de Agronomia, Universidade Estadual de Maringá, Maringá 87020-900, PR, Brazil; djtessmann@uem.br; 5Departamento de Ciência e Tecnologia de Alimentos, Faculdade de Agronomia Eliseu Maciel, Universidade Federal de Pelotas, Capão do Leão 96160-000, RS, Brazil; cesarvrf@ufpel.edu.br; 6Ambev, Passo Fundo 99040-630, RS, Brazil; 99808098@ambev.com.br

**Keywords:** food security, food contamination, *Fusarium* spp., *Hordeum vulgare*, mycotoxins

## Abstract

Fusarium head blight in barley (*Hordeum vulgare*) reduces grain yield and can lead to the accumulation of deoxynivalenol (DON) and nivalenol (NIV) in grains. We surveyed *Fusarium* species and evaluated DON and NIV concentrations in barley grains in four regions of Rio Grande do Sul, the southernmost state in subtropical Brazil. Seven *Fusarium* species were identified: *F. asiaticum*, *F. avenaceum*, *F. cortaderiae*, *F. graminearum*, *F. gerlachii*, *F. meridionale* and *F. poae*. DON (0 to 10,200 µg/kg) and NIV (0 to 1630 µg/kg) were detected in 74% and 70% of the samples, respectively, with higher concentrations found in experimental fields. However, in commercial barley fields, most samples fell below 2000 µg/kg of DON, which is the maximum limit allowed by Brazilian legislation for grains intended for processing. The seasonality of temperature and precipitation influenced mycotoxin concentrations. Therefore, the variability of *Fusarium* species in Rio Grande do Sul and a high incidence of DON and NIV in barley grains highlight the complexity of this pathosystem. This variability of *Fusarium* species may also influence the effectiveness of measures to control the disease, particularly in relation to genetic resistance and fungicide application.

## 1. Introduction

Barley (*Hordeum vulgare*) is a cereal grown worldwide due to its adaptability to different environments and the multifunctional uses of its grains. In Brazil, barley is mainly used in the brewing industry. Two-row spring barley is grown mainly in the southern region, where the states of Paraná and Rio Grande do Sul are the largest producers [1]. In Rio Grande do Sul, the southernmost state, barley is cultivated in physiographic regions differing in climate such as temperature and humidity, which influence the occurrence of diseases as well as yield and grain quality.

Fusarium head blight (FHB), caused by species of the genus *Fusarium*, is one of the main fungal diseases affecting barley worldwide. It reduces grain yield and quality and leads to the accumulation of mycotoxins in infected grains [2,3,4]. Among the mycotoxins produced by *Fusarium* species, trichothecenes stand out for their phytotoxic effects on plants and their harmful effects on human and animal health [5]. Type B trichothecenes include the mycotoxins deoxynivalenol (DON), its acetylated derivatives 3-acetyldeoxynivalenol (3ADON) and 15-acetyldeoxynivalenol (15ADON) and nivalenol (NIV).

DON is known to cause food refusal, vomiting and weight loss in animals. In humans, the effects of DON include anorexia, immunotoxicity, diarrhea, vomiting, leukocytosis, circulatory shock and even death [5]. In turn, the toxic effects of NIV are not yet clearly understood, but studies have shown that DON and NIV share similar toxic effects. Notably, in both in vivo and in vitro assays, NIV has been found to have a greater negative impact on pig intestinal mucosal cells than DON [6]. Furthermore, even at low concentrations, consuming foods contaminated with NIV can have harmful effects on human health. In vitro studies have shown that human cells can be more sensitive to NIV than animal cells, suggesting potentially greater damage in humans. Additionally, there may be a synergistic effect of DON and NIV [7].

Due to the significant risk posed by mycotoxins on food, several countries have established acceptable limits for their concentration in cereal grains. For example, the maximum tolerated limit (MTL) for DON in grains used for brewing is set at 2000 µg/Kg in Brazil, while it is 500 µg/Kg in the European Union and Canada [8,9]. To the best of our knowledge, no country has established a MTL for NIV, which poses a food safety risk due to the potential health harm caused by ingesting this mycotoxin.

The type and concentration of mycotoxins in grains vary between fields depending on the *Fusarium* species present, which can be influenced by agronomic practices and the physiographic region [10,11]. Furthermore, the occurrence and prevalence of *Fusarium* species in a given region can be altered by climate changes, as highlighted in a previous study [12]. This makes continuous monitoring of *Fusarium* species a vital step to control FHB and consequently reduce grain contamination by mycotoxins.

Surveys of *Fusarium* species across various cereal fields worldwide have revealed significant variation. In France, the leading barley-producing country in the European Union, the most commonly found species in barley and wheat (*Triticum aestivum*) grains are *F. graminearum*, *F. tricinctum*, *F. avenaceum* and *F. poae* [13]. In Canada, *F. poae*, *F. graminearum* and *F. avenaceum* are the predominant species [14]. In the United States, *F. graminearum sensu stricto* (*s.str*.) is mainly responsible for the occurrence of FHB in barley and wheat plants, though an increase in the occurrence of *Fusarium tricinctum* species complex has also been reported [2,10]. In China, *F. avenaceum* was the prevalent species, but *F. equiseti*, *F. verticillioides*, *F. acuminatum*, *F. flocciferum* and *F. proliferatum* were also found [15]. In Brazil, particularly the state of Paraná, *F. graminearum s.str.* is recognized as the primary species responsible for causing FHB in barley. However, other species, such as *F. austroamericanum* and *F. meridionale*, have also been identified as causal agents [16].

In Brazil, the state of Rio Grande do Sul is the second-largest barley producer, and it can be divided into eleven physiographic regions [17]. Among these, the physiographic regions of Alto Uruguai, Campos de Cima da Serra, Encosta do Sudeste and Planalto Médio stand out as the main barley-producing areas. Altitude, temperature and precipitation are important factors that differentiate these regions. However, to the best of our knowledge, information about *Fusarium* species affecting barley and the occurrence of mycotoxins is scarce for these regions. In view of the above, our hypothesis was as follows: (i) different *Fusarium* species are responsible for FHB in barley across distinct regions of Rio Grande do Sul; (ii) DON and NIV concentrations vary due to the presence of specific *Fusarium* species; and (iii) environmental conditions, particularly temperature, rainfall and humidity, affect the accumulation of DON and NIV in barley.

To test these hypotheses, we conducted a comprehensive study with the following specific objectives: (i) to identify the *Fusarium* species associated with Fusarium head blight (FHB) in barley from the main barley-producing regions of Rio Grande do Sul, Brazil, by collecting symptomatic barley spikes in ten municipalities across four physiographic regions; (ii) to detect and quantify the concentrations of the mycotoxins deoxynivalenol (DON) and nivalenol (NIV) in grain samples collected from fields in 11 municipalities, mostly corresponding to the same locations where spikes were sampled; and (iii) to evaluate the influence of bioclimatic variables—particularly temperature, rainfall and humidity during the year—on DON and NIV concentrations in barley grains.

## 2. Material and Methods

### 2.1. Barley Sampling

Samples of barley spikes and grains (0.5 to 1.0 kg) from the 2021 crop season were collected in commercial fields and in experimental areas of the Federal University of Pelotas (UFPel) and the beverage company AMBEV (Table 1). The sampled areas belong to four physiographic regions of Rio Grande do Sul (Figure 1), namely Alto Uruguai (municipalities of Estação and Ipiranga do Sul); Campos de Cima da Serra (municipalities of Lagoa Vermelha and Vacaria); Encosta do Sudeste (municipality of Capão do Leão); and Planalto Médio (municipalities of Água Santa, Coxilha, Passo Fundo, Soledade, Tapejara and Vila Lângaro). These regions vary in altitude as well as mean temperatures and rainfall levels [17].

The samples included five barley genotypes, four of which were cultivars (‘Ana 02’, ‘BRS Brau’, ‘BRS Cauê’ and ‘ABI Rubi’) and one a lineage (0078). The cultivars sampled were the main ones used by barley growers. In all sampled areas, fungicides were applied to control FHB, but some samples from the municipality of Capão do Leão were obtained from experimental fields without fungicide treatment. The samples from AMBEV came from experimental fields located in the municipalities of Estação, Vacaria, Agua Santa, Coxilha, Passo Fundo and Soledade where the same agronomic management and barley genotypes were employed in all fields. Commercial fields received different disease management, specifically by employing different fungicides (Appendix A), while the remaining agronomic management was similar among fields.

For the spikes, 16 fields were sampled, and in 8 of these fields, more than one barley genotype was cultivated (Table 1). For each sampled cultivar, at least 20 symptomatic spikes (at phenological stages ranging from 83 to 87 on the Zadoks scale) were randomly collected from different points within each cultivated field. The spikes were transported to the laboratory, identified and stored at 4 °C for up to two days. For the grains, 42 samples were obtained during machine harvesting at physiological maturity. The grains were then dried in the laboratory until they reached 13% moisture content. Afterward, they were packed in paper bags and stored in a refrigerator at 4 °C until analysis. In this study, a sample was considered to be the pool of spikes (~20 spikes) or the grains (0.5 to 1.0 kg) obtained from a specific cultivar in each sampled field.

### 2.2. Pathogen Isolation from Symptomatic Barley Spikes

Barley spikes displaying FHB symptoms (symptoms confined to a single spikelet, affecting the entire spike, symptomatic spikes still enclosed within the flag leaf sheath, or those exhibiting color variations) were selected for fungal isolation. For spikes exhibiting visible signs of the pathogen, such as vegetative or reproductive fungal structure observable to the naked eye, direct isolation was performed under a stereomicroscope in an aseptic environment. The direct fungal isolation consisted of the transfer of conidia from the spike surface to Petri dishes containing potato–carrot agar (PCA) culture medium. For spikes without visible signs of the pathogen, a humid chamber was used to externalize the pathogen structures, from which the direct pathogen isolation was performed. To create the humid chamber, spikes with FHB symptoms were placed inside a gerbox (JProlab, SP, Brazil) (11 × 11 × 3.5 cm), over filter paper moistened with sterile water, ensuring there was no direct contact between the spike and the filter paper and incubated in a BOD incubator at a temperature of 25 °C with a 12 h photoperiod for 48 to 72 h. All obtained *Fusarium* isolates were then grown on PCA in a BOD incubator at 25 °C with a 12 h photoperiod.

### 2.3. DNA Extraction, Polymerase Chain Reaction (PCR) and Sequencing

Fungal isolates were grown in Petri dishes containing potato–dextrose agar (PDA) for 10 days at 25 °C and a 12 h photoperiod. Genomic DNA was extracted using the Promega Wizard DNA Extraction kit (Promega Corporation, Madsion, WI, USA), following the manufacturer’s protocol. After extraction, the rehydrated fungal DNA was quantified and its quality assessed using a NanoVue Plus Spectrophotometer (Biochrom, Holliston, MA, USA). The DNA was then stored in a freezer at −20 °C.

Two genomic regions were targeted for amplification by PCR and sequencing. The primer pairs EF-1 (5′-ATGGGTAAGGARGACAAGAC-3′) and EF-2 (5′- GGARGTACCAGTSATCATG-3′) [18] were used for the protein-coding region translation elongation factor 1-alpha (*tef1*). The primer pairs 5F2 (5′ GGGGWGAYCAGAAGAAGGC -3′) [19] and 7cR (5′ CCCATRGCTTGYTTRCCCAT-3′) [20] were used for the second largest subunit of RNA polymerase II (*rpb2*).

Amplifications were performed in a thermocycler (BioRad, Model T100, Hercules, CA, USA) using Promega PCR Master Mix (Promega Corporation, Madsion, WI, USA) (50 units/mL of Taq DNA polymerase, 400 µM dATP, 400 µM dGTP, 400 µM dCTP, 400 µM dTTP and 3 mM MgCl2). The reaction was carried out in a final volume of 25 µL, containing 12.5 µL of the PCR Master Mix, 1 µL of each primer (forward and reverse) at a concentration of 10 µM, 2 µL of DNA (varying in concentrations from 100 to 400 ng/µL) and 8.5 µL of ultrapure water. The PCR conditions for amplification of each region are described in Appendix A. Afterward, the amplified fragments were loaded onto a 1% agarose gel, labeled with GelRed, for gel electrophoresis to visualize DNA fragments. The PCR products were then purified with ReliaPrepTM DNA Clean-up and Concentration System (Promega), following the manufacturer’s protocol and stored in a freezer at −20 °C. The amplified fragments were sequenced in both forward and reverse directions using the same primers as in the PCR, through Sanger sequencing. Consensus sequences were generated by comparing and manually editing the ab1 forward and reverse files, which were visualized with MEGA (Molecular Evolutionary Genetics Analysis) v. 7.0. Sequences were deposited in GenBank and accession numbers are listed in the Appendix A.

### 2.4. Phylogenetic Analysis

Consensus sequences were generated and imported into MEGA 7.0 for initial alignment through Clustal W and database construction. Sequences obtained from the alignment by Crous et al. [21], from GenBank (http://www.ncbi.nlm.nih.gov), along with the new sequences obtained in the present study, were initially aligned using the MAFFT program v.7 [22]. Manual adjustments were made in MEGA 7.0 when necessary. After the initial analysis, the database was refined to include only the isolates under study and their close taxa.

Genetic evolution models were selected using MrModeltest v. 2.3 [23], which identified the GTR+I+G base substitution model as the most appropriate for all gene partitions in the database. Based on the results from MrModeltest v. 2.3, Bayesian inference was conducted using the CIPRES platform [24] with the MrBayes v. 3.2.1 program. For the Bayesian inference, we used *tef1* and *rpb2* genes in a concatenated form. Posterior probability values were determined through Markov chain Monte Carlo (MCMC) sampling. Six simultaneous Markov chains were run for 10,000,000 generations, with phylogenetic trees displayed every 100 generations until convergence was reached (stopval = 0.01). The first trees generated (25% burn-in) were discarded, and the remaining trees were used to calculate posterior probabilities (PP). *Atractium stilbaster* (isolate CBS 410.67) served as the outgroup for the Bayesian analysis, consistent with the outgroup used by Crous et al. [21].

### 2.5. Mycotoxin Quantification

Among the various mycotoxins produced by *Fusarium* species, we selected deoxynivalenol (DON) and nivalenol (NIV) for quantification in barley grains. DON stands out as one of the most important mycotoxins worldwide, with clearly defined maximum tolerance limits in several countries, while NIV has been frequently detected in cereal grains. Therefore, determining the concentrations at which this mycotoxin occurs is crucial to support the development of future regulatory frameworks aimed at preventing intake levels that may pose a risk to human health.

Fifty grams of each sample of barley grains was ground until passing through a 1 mm sieve using an ultracentrifugal mill (ZM 200; Retsch, Hann, Germany). The samples were then homogenized and analyzed for the presence and concentration of mycotoxins by HPLC-MS/MS at the Laboratory of Mycotoxicological Analysis of Federal University of Santa Maria). The chemical reagents used were methanol, acetonitrile, ammonium acetate and acetic acid (all HPLC grade). They were purchased from J.T. Baker (Center Valley, PA, USA). Water was purified using the Milli-Q Direct 8 Water Purification System (Millipore, Burlington, MA, USA).

#### 2.5.1. Deoxynivalenol Analysis

For the analysis of deoxynivalenol (DON), the method proposed by Simões et al. [25] was utilized. A 3 g sample was mixed with a 24 mL methanol/ultrapure water solution (70:30, *v*/*v*) and agitated for 20 min using an orbital shaker (Marconi, MA563, Piracicaba, SP, Brazil). The resulting extract was centrifuged at 1258× *g* at 20 °C for 5 min (Eppendorf, 5804R, Hamburg, HH, Germany). Subsequently, 40 μL of the supernatant was diluted in 960 μL of a methanol/ultrapure water/ammonium acetate solution (90:9:1, *v*/*v*/*v*).

#### 2.5.2. Nivalenol Analysis

NIV was analyzed using the method proposed by Mallmann et al. [26] with adaptations. A 5 g sample was mixed with a 20 mL solution containing acetonitrile and ultrapure water (84:16, *v/v*) and agitated on an analog shaker table (Lucadema, LUCA-180/A, São José do Rio Preto, SP, Brazil) for 1 h. The extract was then centrifuged at 1258× *g* at 20 °C for 5 min, and 10 mL of supernatant was evaporated under a nitrogen flow at 65 °C. The residue was resuspended in a solution of acetonitrile/ultrapure water/acetic acid (840:160:5, *v*/*v*/*v*), shaken for 1 min and passed through a MycoSep^®^ 227 Trich+ column (Romers Lab, Campinas, SP, Brazil). Finally, 20 μL of the eluate was diluted in 980 μL of a methanol/ultrapure water solution (1:1, *v*/*v*).

#### 2.5.3. HPLC-MS/MS Analysis Parameters

A sample of 10 μL was injected into an Agilent 1200 Infinity HPLC system coupled to an API mass spectrometer 5000, equipped with an ESI source, operating in both positive and negative mode. The mobile phase gradient consisted of ultrapure water/ammonium acetate (99:1, *v*/*v*) and methanol/ultrapure water/ammonium acetate (90:9:1, *v*/*v*/*v*). Chromatographic separation was conducted at 40 °C using a Zorbax SB-C8 column (4.6 × 150 mm, 5 μm) from Agilent Technologies Inc. (Santa Clara, CA, USA).

The limit of quantification (LOQ) and limit of detection (LOD) were established using the signal-to-noise ratio (LOQ = 10/1, LOD = 3/1). The LOD and LOQ (in μg/kg) for the assessed mycotoxins were 50 and 200 for DON, and 80 and 100 for NIV, respectively. The linearity of the analytical curves was evaluated through the coefficient of determination (R^2^) from analytical curves obtained from triplicate injections, with the R^2^ values being ≥0.99. Data acquisition was carried out using the Analyst software (1.4.2, Sciex, Framingham, MA, USA).

### 2.6. Data Analysis

In our study, incidence (%) indicates the percentage of the number of samples with a given taxon (species) or specific mycotoxins over the total number of samples. To evaluate the effect of bioclimatic variables on the concentration of DON and NIV, generalized linear models (GLMs) with a normal distribution were constructed. The bioclimatic variables were obtained from the WorldClim database [27] (Table 2) based on the geographic coordinates of each sampled field (Table 1). The variables BIO4 and BIO 15 were fixed based on data from the literature [28,29] and the others were based on the lowest variance inflation factor (VIF < 10.0) values. From the full model, different combinations and interactions among the selected bioclimatic variables were examined. Twenty-five GLMs were generated for each response variable to identify the simplest model by the lowest Akaike information criterion (AIC) value.

To evaluate whether the presence and concentration of DON and NIV could be influenced by the cultivar and the physiographic region, we performed factor analysis with mixed data (FAMD). This is a principal component analysis adapted to analyze datasets with categorical (qualitative) and continuous (quantitative) variables [30]. Data analysis was carried out with the R software version 4.3.2 (R Development Core Team 2023). The FactoMineR package version 2.9 was also used to analyze the dataset, and the Factoextra package version 1.0.7 was used to visualize the data. In these analyses, DON and NIV were used as quantitative variables while physiographic region (PR) and cultivar (CV) were used as qualitative variables.

## 3. Results

### 3.1. Fusarium Species Identification

A total of 150 fungal isolates were obtained from symptomatic samples. Isolates differing in growth rate and colony morphology (mycelial density and color) were selected for molecular analysis, resulting in 70 isolates. The DNA sequences of the *rpb2* and *tef1* genes of those 70 isolates of *Fusarium* spp were used for alignment. The final alignment consisted of 1.519 nucleotides comprising the 70 *Fusarium* isolates from this study plus 31 sequences of *Fusarium* isolates obtained from NCBI (study of Crous et al., [21]) and an outgroup taxon. The alignment comprised 878 nucleotides for the *rpb2* gene and 687 nucleotides for the *tef1* gene. The resulting phylogenetic tree (Figure 2) grouped all isolates into two species complexes, specifically *F. sambucinum* and *F. tricinctum*.

The phylogenetic analysis showed that 31 of the 70 isolates clustered into the same clade with *F. graminearum* having high support (0.98/1). The phylogenetic tree also showed that eight isolates clustered with *F. meridionale* with high support (1.0/1), two isolates clustered with *F. asiaticum* with high support (1.0/1), three isolates clustered with *F. cortaderiae* with high support (0.8/1), eight isolates clustered with *F. gerlachii* with high support (1.0/1), four isolates clustered with *F. poae* with high support (1.0/1) and one isolate (LIPP 56) not grouped with any species included in the present analysis. The phylogenetic tree showed that five of the seventy isolates clustered into the *F. tricinctum* complex, namely three isolates clustered with *F. avenaceum* with high support (1.0/1), while two isolates were clustered at the base of the *F. avenaceum* clade.

*Fusarium graminearum* was detected in almost all sampled municipalities across the four physiographic regions (Table 3). In contrast, *F. asiaticum* was found only in the warmest region (Encosta do Sudeste), where the occurrence of *F. avenaceum* was also observed. *Fusarium cortaderiae* was found in the cooler region (Campos de Cima da Serra), but it also was detected in the region of Planalto Medio, along with three other species: *F. graminearum*, *F. meridionale* and *F. gerlachii* (Table 3). Meanwhile, six species were recorded in both Alto Uruguai and Campos de Cima da Serra: *F. avenaceum*, *F. cortaderiae*, *F. graminearum*, *F. gerlachii*, *F. meridionale* and *F. poae*. In samples from Campos de Cima da Serra, one isolate could not be identified at the species level (Table 3).

### 3.2. Detection and Quantification of DON and NIV

The mycotoxins DON and NIV were found in 74 and 70% of the samples, respectively (Table 4). In general, the concentration of DON in barley grains was higher than that of NIV. DON concentration ranged from zero (0) to 10,200 µg/Kg, while NIV concentration ranged from zero (0) to 1630 µg/Kg. In commercial fields, DON and NIV were detected in almost all samples, but only one sample had a DON concentration above the LMT (2000 µg/Kg) (Table 4). In experimental areas where the same experiment was conducted, DON was detected in 87% of the samples, with 33% having concentration above the Brazilian LMT. NIV was detected in 80% of the samples, with concentration ranging from 62.7 to 595 µg/Kg (Table 4).

In the municipality of Capão de Leão, the highest concentrations of both DON and NIV were detected in grains from area 1, while in the area 2, only NIV was detected in two samples, with concentration below 135 µg/Kg (Table 4). In area 1, the highest concentration of DON occurred in plants not treated with fungicide, while fungicide treatment for FHB control reduced the DON concentration to about half that of untreated plants (Table 4). On the other hand, the same fungicidal treatment was not observed for NIV.

In contrast, in samples collected in Vacaria (Campos de Cima da Serra region), mycotoxins were either not detected or detected in low concentrations, except in one sample, in which the concentration of DON was 1960 µg/Kg (Table 4).

Regarding the influence of bioclimatic variables on mycotoxin concentrations, we identified that temperature seasonality (BIO4) and precipitation seasonality (BIO15) were important to both DON and NIV (Table 5).

#### Concentrations of DON and NIV by Cultivar and Physiographic Region

In this analysis, we excluded the Encosta do Sudeste region because only two cultivars were sown, and for one of them there was only one field. To ensure greater reliability of the results by cultivar and physiographic region, we only considered fields where the same cultivars were sown and identical agronomic practices were adopted. Regarding genotypes, no significant differences were found among them for both DON and NIV concentrations (Figure 3A,B). However, for municipalities, barley plants from Vacaria showed lower DON and NIV concentration than those from Água Santa, Coxilha, Passo Fundo, Soledade and Estação (Figure 3C,D). These results indicated that the fields from the physiographic region of Campos de Cima da Serra were less favorable to DON and NIV accumulation in grains compared to Alto Uruguai and Planalto Medio (Figure 3E,F) during the crop season of 2021.

The results of a factor analysis of mixed data (FAMD) applied to mycotoxin data are presented in Figure 4. The figure showing mycotoxins by physiographic region (PR) displays the distribution of samples based on the first two dimensions (Dim1: 52.2%, Dim2: 25%) (Figure 4A). The samples are grouped according to the variable PR into three categories, namely Alto Uruguai (red), Campos de Cima da Serra (blue) and Planalto Médio (green). The clustering indicates differences in mycotoxin profiles among the categories. The vector plot shows the contribution of two mycotoxins (NIV and DON) to the first two dimensions (Figure 4B). Both vectors point towards dimension 1, suggesting a strong association of these mycotoxins with the first principal component.

Regarding physiographic region, both mycotoxins tend to vary similarly, as indicated by the vector graph, which means that when the environment is favorable for DON, it is also conducive to NIV accumulation. This result is also supported by positive correlation (0.49) between DON and NIV concentrations (Appendix A). Regarding the sample distribution by the cultivars (CV), they were grouped into four categories: Ana 2 (red), BRS Caue (green), Lineage 0078 (yellow) and Rubi (purple) (Figure 4C). The clustering patterns reveal variability in mycotoxin profiles across the different groups. The vector graph resembles the interpretation to the PR with NIV and DON contributing predominantly to Dim1 (33%) and Dim2 (21%) (Figure 4D). In general, the clustering of samples indicates that the mycotoxin profiles differed according to the categorical variables PR and CV. Although both mycotoxins tended to vary similarly regarding cultivar, NIV showed positive scores in both dimensions, while DON showed a negative score in dimension 2, indicating that mycotoxins may also be influenced by barley genotype, depending on the environmental conditions.

## 4. Discussion

This study revealed that the occurrence of FHB in barley plants from different physiographic regions of Rio Grande do Sul, Brazil, is caused by at least seven species of *Fusarium*. These species are influenced to some extent by the intrinsic conditions of each region, mainly variations in temperature and precipitation, which, along with the *Fusarium* species present, may have affected the mycotoxin concentrations. Furthermore, it demonstrated that up to three *Fusarium* species may co-exist within the same field and/or municipality, which poses a challenge to disease management.

An interesting fact is that a previous study carried out in Rio Grande do Sul during the period from 2007 to 2009 identified *F. graminearum* and *F. meridionale* as the species associated with FHB in barley [31]. In the current survey, we identified five additional species: *F. asiaticum*, *F. avenaceum*, *F. cortaderidae*, *F. gerlachii* and *F. poae*. The differences in the method used to identify the *Fusarium* species between the two studies may explain the increase in the number of species associated with FHB in barley. However, another study performed with wheat in two states of the southern region of Brazil found five species: *F. graminearum*, *F. meridionale F. asiaticum*, *F. cortaderidae* and *F. austroamericanum*, with *F. asiaticum* and *F. cortaderidae* only being found in Rio Grande do Sul [32]. Therefore, it is reasonable to consider that an increase in the diversity of *Fusarium* species may have occurred in barley crops, a situation that requires deeper studies to shed light on more accurate management’s strategies, especially in a changing climatic environment.

*Fusarium graminearum* was the predominant species, comprising around 50% of the isolates obtained in the study and distributed across all physiographic regions. Similarly, a previous survey carried out in wheat and barley fields in the state of Paraná also identified *F. graminearum* as the predominant species [16], as well as a recent study carried out in Argentina [33]. Interestingly, in the spikes collected in Água Santa, Coxilha and Vila Lângaro, where all isolates were grouped with *F. graminearum*, and in Capão do Leão (Area 1), where *F. graminearum* was the main species, the highest concentrations of DON were observed in the grains, an indication that this species contributes substantially to DON accumulation. On the other hand, *F. asiaticum* was only detected in Encosta do Sudeste, likely indicating a more restricted distribution of this species, possibly associated with more specific bioclimatic variables required. Other *Fusarium* species that were recorded in at least three physiographic regions were *F. avenaceum, F. cortaderiae*, *F. gerlachii* and *F. meridionale*, indicating that these species were also widely distributed across the state. Interestingly, *F. gerlachii* was not reported in previous studies [16,31,32], while we did not find *F. austroamericanum* in the current study. This result may be due to differences in sampled fields but may also indicate dynamic changes in less frequent species in the cereal fields.

Campos de Cima da Serra was the region with the highest diversity of *Fusarium* species, but it was also where, in general, the lowest concentrations of mycotoxins were detected. This result is probably associated with the bioclimatic variables, which influence the accumulation of mycotoxins. Since *F. graminearum* was the predominant species in all regions and primarily produces DON, this may explain the higher concentration of this mycotoxin in the grains. A previous study demonstrated that in co-inoculated samples with different *Fusarium* species, *F. graminearum*-associated metabolites dominated the mycotoxin profile [34], probably because it is favored during warm nights [35]. This result is corroborated by the highest concentration of DON and NIV in Capão do Leão (Area 1). Furthermore, the co-occurrence of more than one *Fusarium* species in the same field may favor multi-mycotoxin contamination alongside changing temporal–geographical distributions.

Interestingly, DON and NIV occurred concomitantly in 71% of the samples. Previous studies have reported the co-occurrence of mycotoxins in barley and wheat fields in southern Brazil [36] and in other barley-producing countries, such as Canada [14] and Spain [37]. In most cases, DON concentrations were higher than those of NIV. Similar results were reported in previous studies carried out in Brazil [38], Argentina [31], Spain [37] and the Czech Republic [39]. This variance may be related to differences in the pathogen’s adaptability, as well as to *Fusarium* genotypes that produce DON, such as *F. graminearum,* which was found in all regions. This corroborates a previous study that found it in greater frequency in cereal fields in the southern region of Brazil than genotypes that produce NIV [31]. Furthermore, DON production is favored by higher temperatures (25 to 28 °C) and water activity ranging from 0.96 to 0.98, while NIV production is favored by lower temperatures, around 20 °C, and high water activity (0.98) [40,41,42]. These specific requirements for each mycotoxin are consistent with our observation of the positive effect of temperature and rainfall seasonality, correlated with the variation in mycotoxin concentrations among the physiographic regions studied.

The concentrations of DON observed in the present study were comparable to those previously reported in Uruguay [43] but exceeded the levels documented in earlier studies conducted in Brazil [44] and Argentina [33]. In contrast, the concentrations of NIV were consistent with those reported in both Argentina and Brazil. These variations in DON levels may be attributed to differences in Fusarium species composition, agronomic practices and environmental conditions among regions and cropping seasons.

In the sampled areas, the Encosta do Sudeste region had the highest average annual temperature and the lowest altitude among the studied regions. The highest concentrations of DON and NIV were detected in this region in all samples analyzed. In Encosta do Sudeste, especially area 1, the reproductive stage of the crop coincided with a period of frequent rainfall, so this factor combined with high temperatures likely provided optimal conditions for the development of the pathogen and consequently contributed to the secretion of mycotoxins in high concentrations. Of particular note, for this same region the non-coincidence of the emergence/flowering of barley plants with rainy periods prevented the occurrence of FHB and the contamination of grains by DON and NIV, as occurred in area 2. However, in this area, a predominance of isolates grouped with *F*. *avenaceum* was also found, for which trichothecene production is not known [45]. This result indicates that even for regions very favorable to the disease and the accumulation of mycotoxins, bioclimatic variables can be favorable to the crop and allow the production of barley grains of acceptable quality. On the other hand, in the region of Campos de Cima da Serra, the sampled areas were located at altitudes above 900 m, in which the average annual temperature was the lowest among the physiographic regions. In this region, in general the samples presented low DON concentrations, with some samples having no detectable DON or NIV.

The regions of Alto Uruguay and Planalto Médio, in which altitude, temperature and average rainfall were similar, the Alto Uruguay region in general presented lower concentrations of DON and NIV. Although meteorological factors during the reproductive phase may have influenced this result, the difference in the population of *Fusarium* species, with prevalence or *F. graminearum*, may also have been a factor that contributed to the variations in concentrations between the two regions.

Because of the high occurrence of DON and NIV in samples collected in regions with variations in altitude, temperature and relative humidity, it is important to reinforce the monitoring of mycotoxins in barley grains as well as to develop or adapt measures to mitigate this problem. The NIV occurrence is particularly worrying given the increasing observations of its negative effects on the health of humans and animals, and the frequence it was found in barley grains. This study focused exclusively on the analysis of DON and NIV, both of which are type B trichothecenes. Future studies could investigate the presence of type A trichothecenes, such as enniatins, potentially produced by *F. avenaceum,* among other mycotoxins potentially produced by *Fusarium* species [45] that were found in this study. Another important observation is that genetic materials (cultivar and lineage) under different soil and climate conditions had similar behavior, with little or no influence on the concentration of mycotoxins.

Finally, the high number of *Fusarium* species associated with FHB in barley highlights the need for further studies to identify the genetic response of cultivars to these pathogens, as well as to verify whether there is variable sensitivity to fungicides among species. An eventual shift in species composition resulting from climatic factors or cultural practices also deserves further investigation. Such information is essential for developing strategies to improve FHB control and reduce mycotoxin contamination in grains.

## Figures and Tables

**Figure 1 plants-14-02327-f001:**
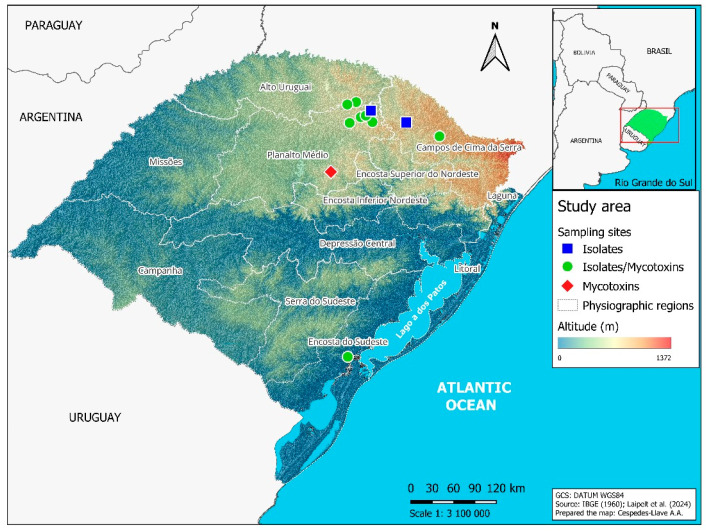
Map of Rio Grande do Sul state in Brazil, indicating the municipalities where barley samples were collected during the 2021 crop season. Blue squares represent municipalities where only barley spikes were sampled for fungal isolate identification. Green circles indicate municipalities where spikes were collected for fungal identification and grains for mycotoxin quantification. The red diamond marks the municipality where only grains were collected to quantify mycotoxins.

**Figure 2 plants-14-02327-f002:**
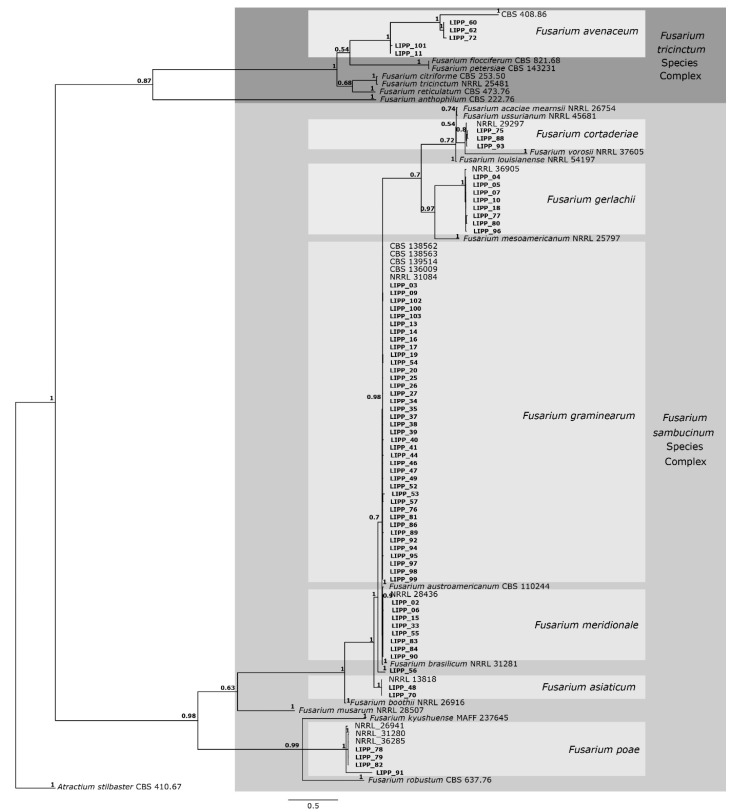
Consensus tree of *Fusarium* species, based on Bayesian analysis of the *rpb2* and *tef1* gene alignment. Posterior probability (PP) values are indicated on the left side of each node. The scale bar indicates 0.5 expected changes per site. Isolates derived from the present study are indicated in bold. The tree is rooted in *Atractium stilbaster* (isolate CBS 410.67).

**Figure 3 plants-14-02327-f003:**
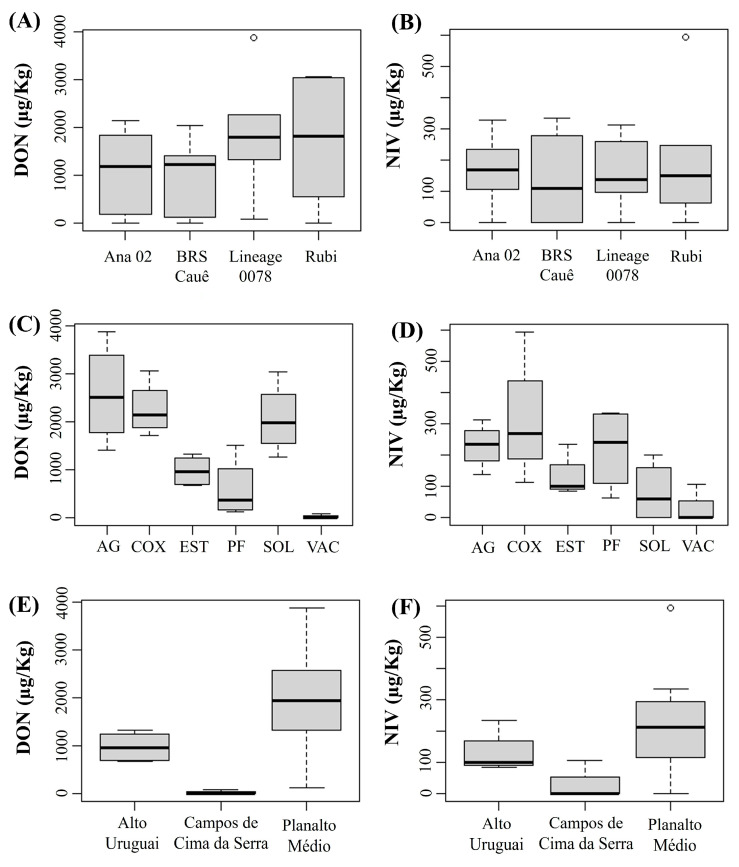
Concentration of deoxynivalenol (DON) (**A**,**C**,**E**) and nivalenol (NIV) (**B**,**D**,**F**) in barley grains of each cultivar sampled (**A**,**B**) in different municipalities (Ag: Água Santa, COX: Coxilha, EST: Estação; PF: Passo Fundo, SOL: Soledade, VAC: Vacaria) (**C**,**D**) of three physiographic regions (**E**,**F**) of the state of Rio Grande do Sul during the 2021 crop season. (Number of samples per cultivar: ‘Ana 02’ = 7, ‘BRS Cauê’ = 19, ‘Lineage 0078’ = 6, ‘ABI Rubi’ = 7). The thick horizontal line inside the box represents the median, the limits of the box represent the lower and upper quartiles, and the circles represent extremes.

**Figure 4 plants-14-02327-f004:**
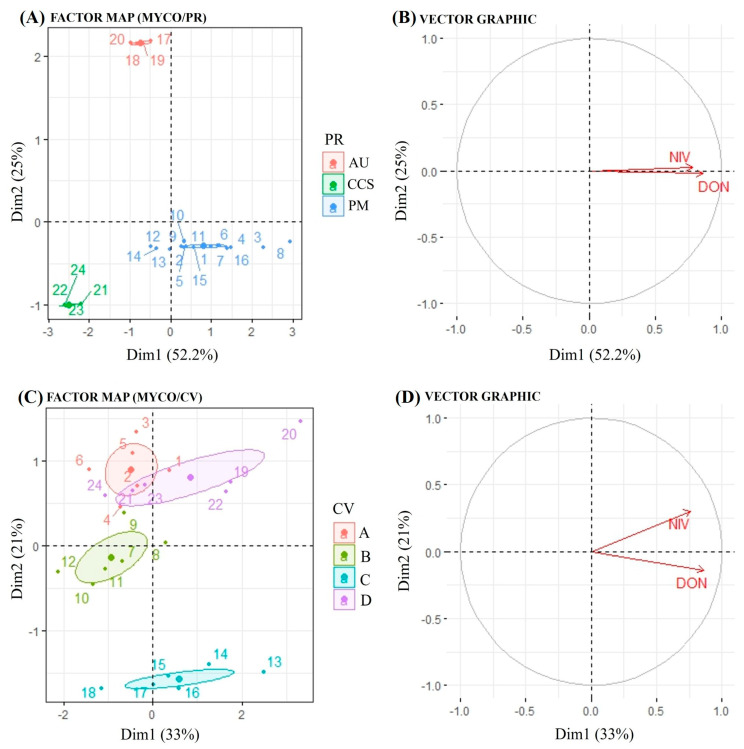
Biplots of factor map and vector graph, obtained from factor analysis of mixed data (FAMD). (**A**) Factor map of mycotoxins and physiographic region; (**B**) vector graph of mycotoxins and physiographic region; (**C**) factor map of mycotoxins and cultivar; (**D**) vector graph of mycotoxins and cultivar. Cultivars (CV): A: ‘Ana 02’/B: ‘BRS Cauê’/C: Lineage 0078/D: ‘ABI Rubi’. Physiographic regions (PR): AU: Alto Uruguai/CCS: Campos de Cima da Serra/PM: Planalto Médio. DON: deoxynivalenol/NIV: nivalenol. MYCO: Mycotoxins DON + NIV.

**Table 1 plants-14-02327-t001:** Physiographic regions, municipality, field designation, geographic coordinates, altitude, barley genotype sampled in the study and type of sample obtained.

Physiographic Regions	Municipality	Field ^1^	Geographic Coordinates		Altitude (m)	Barley Genotype ^2^	Sample	
			Longitude	Latitude			Spike	Grains
Alto Uruguai	Estação	AMBEV	−52.291794	−27.898500	763	A/B/C/D	×	×
		Commercial field	−52.291794	−27.898500	763	B	×	×
	Ipiranga do Sul	Commercial field I	−52.450942	−27.945414	650	B	×	×
		Commercial field II	−52.450942	−27.945414	650	B	×	×
Encosta do Sudeste	Capão do Leão	UFPel Area I	−52.419113	−31.798330	20	B	×	×
		UFPel Area II	−52.505765	−31.800805	20	B/E	×	×
Planalto Médio	Água Santa	AMBEV	−52.045521	−28.191851	794	A/B/C/D	×	×
	Coxilha	AMBEV	−52.215816	−28.093141	627	A/B/C/D	×	×
		Commercial field	−52.24083	−28.108747		D	×	ns
	Passo Fundo	AMBEV	−52.392713	−28.218805	661	A/B/C/D	×	×
	Soledade	AMBEV	−52.677361	−28.969638	697	A/B/C/D	ns	×
	Tapejara	Commercial field	−52.065267	−28.032335		B	×	ns
	Vila Lângaro	Commercial field	−52.123922	−28.107910	643	B	×	×
Campos de Cima da Serra	Lagoa Vermelha	Commercial field	−51.5262	−28.2107		B	×	ns
	Vacaria	AMBEV	−51.016944	−28.423889	962	A/B/C/D	×	×
		Commercial field I	−51.028333	−28.124722	962	A/D	×	×
		Commercial field II	−50.930311	−28.510093	962	B	×	×

^1^ AMBEV and UFPel are experimental fields; commercial fields are those managed by farmers. ^2^ A: cultivar ‘Ana 02’; B: cultivar ‘BRS Cauê’; C: lineage 0078; D: cultivar ‘ABI Rubi’; E: cultivar ‘BRS Brau’; ns: not sampled.

**Table 2 plants-14-02327-t002:** Bioclimatic variables extracted from the WordClim dataset (www.worldclim.org/bioclim) used in distribution models of GLMs for DON and NIV.

Bioclimatic Variables	Descriptions	Unit
BIO1	Annual mean temperature	°C
BIO2	Mean diurnal range (mean of monthly (maximum temperature − minimum temperature))	°C
BIO3	Isothermality (BIO2/BIO7) (∗100)	
BIO4	Temperature seasonality (standard deviation of temperatures ∗ 100)	
BIO5	Maximum temperature of warmest month	°C
BIO6	Minimum temperature of coldest month	°C
BIO7	Annual temperature range (BIO5-BIO6)	°C
BIO8	Mean temperature of wettest quarter	°C
BIO9	Mean temperature of driest quarter	°C
BIO10	Mean temperature of warmest quarter	°C
BIO11	Mean temperature of coldest quarter	°C
BIO12	Annual preciptation	mm
BIO13	Precipitation of wettest month	mm
BIO14	Precipitation of driest month	mm
BIO15	Precipitation seasonality (Coefficient of variation)	mm
BIO16	Precipitation of wettest quarter	mm
BIO17	Precipitation of driest quarter	mm
BIO18	Precipitation of warmest quarter	mm
BIO19	Precipitation of coldest quarter	mm

**Table 3 plants-14-02327-t003:** Identification of the isolates by fungal species according to phylogenetic analysis, municipalities and physiographic region, and type of field where each sample was obtained.

Fusarium Species	Isolates	Municipalities	Physiographic Region	Field
*Fusarium asiaticum*	LIPP 48	Capão do Leão	Encosta do Sudeste	UFPel I
	LIPP 70	Capão do Leão	Encosta do Sudeste	UFPel II
*Fusarium avenaceum*	LIPP 60, LIPP 62 and LIPP 72	Capão do Leão	Encosta do Sudeste	UFPel II
	LIPP 11	Ipiranga do Sul	Alto Uruguai	Commercial field II
	LIPP 101	Lagoa Vermelha	Campos de Cima da Serra	Commercial field
*Fusarium cortaderiae*	LIPP 75	Estação	Alto Uruguai	AMBEV
	LIPP 88	Vacaria	Campos de Cima da Serra	AMBEV
	LIPP 93	Coxilha	Planalto Médio	Commercial field
*Fusarium gerlachii*	LIPP 77 and LIPP 80	Estação	Alto Uruguai	AMBEV
	LIPP 96	Lagoa Vermelha	Campos de Cima da Serra	Commercial field
	LIPP 18	Tapejara	Planalto Médio	Commercial field
	LIPP 04, LIPP 05, LIPP 07 and LIPP 10	Passo Fundo	Planalto Médio	AMBEV
*Fusarium graminearum s.str.*	LIPP 46, LIPP 47, LIPP 53, LIPP 95, LIPP 99, LIPP 100 and LIPP 102	Água Santa	Planalto Médio	AMBEV
	LIPP 25, LIPP 26, LIPP 27, LIPP 49 and LIPP 52	Capão do Leão	Encosta do Sudeste	UFPel I
	LIPP 37, LIPP 38, LIPP 39, LIPP 40, LIPP 86, and LIPP92	Coxilha	Planalto Médio	AMBEV
	LIPP 41, LIPP 44, and LIPP54	Coxilha	Planalto Médio	Commercial field
	LIPP 16, LIPP 34, LIPP 76, LIPP 94 and LIPP 103	Estação	Alto Uruguai	AMBEV
	LIPP 13 and LIPP 14	Ipiranga do Sul	Alto Uruguai	Commercial field I
	LIPP 03 and LIPP 09	Passo Fundo	Planalto Médio	AMBEV
	LIPP 17 and LIPP 35	Tapejara	Planalto Médio	Commercial field
	LIPP 81 and LIPP 89	Vacaria	Campos de Cima da Serra	AMBEV
	LIPP 57, LIPP 97 and LIPP 98	Vacaria	Campos de Cima da Serra	Commercial field I
	LIPP 19 and LIPP 20	Vila Lângaro	Planalto Médio	Commercial field *
*Fusarium meridionale*	LIPP 02, LIPP 5 and LIPP 6	Passo Fundo	Planalto Médio	AMBEV
	LIPP 33	Vacaria	Campos de Cima da Serra	Commercial field II
	LIPP 83, LIPP 84 and LIPP 90	Vacaria	Campos de Cima da Serra	AMBEV
	LIPP 15	Estação	Alto Uruguai	AMBEV
*Fusarium poae*	LIPP 78 and LIPP 79	Estação	Alto Uruguai	Commercial field
	LIPP 82	Vacaria	Campos de Cima da Serra	AMBEV
	LIPP 91	Vacaria	Campos de Cima da Serra	Commercial field I
*Fusarium* sp.	LIPP 56	Vacaria	Campos de Cima da Serra	Commercial field II

* Field with DON concentration above the maximum tolerate limit according to Brazilian legislation.

**Table 4 plants-14-02327-t004:** Concentration of the mycotoxins deoxynivalenol (DON) and nivalenol (NIV) in samples of barley grains obtained from different genotypes collected in municipalities belonging to different physiographic regions of Rio Grande do Sul, Brazil.

Physiographic Regions	Municipality	Cultivar	Mycotoxins (µg/Kg)	
			DON	NIV
Alto Uruguai	Estação (AMBEV ^1^)	‘Ana 02’	671	236
		‘BRS Cauê’	1170	82.7
		Lineage 0078	1330	95.7
		‘ABI Rubi’	727	102
	Estação (Commercial field ^2^)	‘BRS Cauê’	381	327
	Ipiranga do Sul (Commercial field I)	‘BRS Cauê’	973	100
	Ipiranga do Sul (Commercial field II)	‘BRS Cauê’	314	166
Encosta do Sudeste	Capão do Leão (UFPEL ^3^ Área I)	‘BRS Cauê’(+F *)	5930	825
		‘BRS Cauê’(−F *)	7620	1540
		‘BRS Cauê’(+F)	4410	1560
		‘BRS Cauê’(−F)	10,200	1630
	Capão do Leão (UFPEL Área II)	‘BRS Cauê’(+F)	0	0
		‘BRS Cauê’(+F *)	0	135
		‘BRS Cauê’(−F *)	0	114
		‘BRS Cauê’(−F)	0	0
		‘BRS Brau’(+F)	0	0
		‘BRS Brau’(+F *)	0	0
		‘BRS Brau’(−F *)	0	0
		‘BRS Brau’(−F)	0	0
Planalto Médio	Água Santa (AMBEV)	‘Ana 02’	2140	222
		‘BRS Cauê’	1400	139
		Lineage 0078	3880	313
		‘ABI Rubi’	2890	246
	Coxilha (AMBEV)	‘Ana 02’	1710	113
		‘BRS Cauê’	2040	278
		Lineage 0078	2260	260
		‘ABI Rubi’	3060	595
	Passo Fundo (AMBEV)	‘Ana 02’	184	327
		‘BRS Cauê’	126	335
		Lineage 0078	1500	154
		‘ABI Rubi’	558	62.7
	Soledade (AMBEV)	‘Ana 02’	1840	0
		‘BRS Cauê’	1270	0
		Lineage 0078	2110	119
		‘ABI Rubi’	3030	201
	Vila Lângaro (Commercial field)	‘BRS Cauê’	2710	334
Campos de Cima da Serra	Vacaria (AMBEV)	‘Ana 02’	0	106
		‘BRS Cauê’	0	0
		Lineage 0078	73.7	0
		‘ABI Rubi’	0	0
	Vacaria (Commercial field I)	‘Ana 02’	1960	0
		‘ABI Rubi’	188	0
	Vacaria (Commercial field II)	‘BRS Cauê’	425	129

Samples obtained in Capão do Leão were exposed to different treatments: +F and −F indicate presence and absence, respectively, of fungicide application for FHB control; * indicates samples obtained from plants grown in soil supplied with calcium silicate. ^1^ Sites indicated as AMBEV mean grain samples obtained in the experimental area of the company in which the same agronomic procedures were adopted. ^2^ Commercial field indicated grain samples obtained from barley growers. ^3^ UFPel (area 1 and 2) mean grains obtained in the Centro Agropecuário da Palma, the experimental field of Federal University of Pelotas.

**Table 5 plants-14-02327-t005:** Generalized linear models (GLMs) used to evaluate the influence of bioclimatic variables on the concentration of DON and NIV.

Mycotoxin	GLM	Model	Family	AIC	VIF
DON	DON~BIO4 + BIO15	DON~ + BIO4 ***** + BIO15	Gaussian	591.6724	1.08; 1.08
NIV	NIV~BIO4 + BIO15	NIV~ + BIO4 + BIO15 .	Gaussian	465.2927	1.08; 1.08

BIO4 (temperature seasonality) and BIO15 (precipitation seasonality (coefficient of variation)). Akaike information criterion (AIC), variance inflation factor (VIF). (+) positive effect (*) *p* > 0.01 and < 0.05; (.) *p* > 0.05 and < 0.1.

## Data Availability

Data are available upon reasonable request to the corresponding author.

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
