# Peer review of "Fusarium Head Blight in Barley from Subtropical Southern Brazil: Associated Fusarium Species and Grain Contamination Levels of Deoxynivalenol and Nivalenol"

_plants, 2025, doi:10.3390/plants14152327_

Round 1
Reviewer 1 Report
Comments and Suggestions for Authors
The study presents valuable data on Fusarium species and mycotoxin DON and NIV concentrations in barley across four regions, including the influence of bioclimatic variables. The manuscript is well written, with a detailed methodology and a clear presentation of the results. However, certain aspects of the paper could be improved to strengthen the overall quality and clarity of the study.
Title and Formatting:
- The genus name Fusarium should be italicized in the title, line 21, and consistently throughout the manuscript.
- A thorough check of the entire text for formatting consistency and typographical issues is recommended.
References:
- Consider adding more recent references, particularly from the last 5–10 years, to better position this study within the context of current research.
Introduction:
- It is unclear why only deoxynivalenol (DON) and nivalenol (NIV), among DON metabolites, were selected for analysis. Please clarify the rationale for focusing on these two.
- The novelty of the study should be more clearly articulated in relation to existing literature. How does this study differ from or advance previous findings?
Methodology:
- Was a matrix effect evaluation in mycotoxin analysis conducted? If so, please include this information.
- During method validation for mycotoxin determination, both repeatability and recovery should be assessed and reported to ensure the accuracy and reliability of the results.
Results and Discussion:
- It would strengthen the discussion to compare the obtained mycotoxin concentrations (DON and NIV) with those reported in previous studies.
Author Response
Dear Reviewer,
We sincerely thank you for your valuable comments and constructive suggestions regarding our manuscript.
Title and Formatting:
- The genus name Fusarium should be italicized in the title, line 21, and consistently throughout the manuscript.
- Author response: Adjustments were made to the title and throughout the manuscript. The term 'Fusarium' in 'Fusarium head blight' was not italicized, as it refers to the common name of the disease rather than the genus name
- A thorough check of the entire text for formatting consistency and typographical issues is recommended.
- Author response: This verification was carried out.
References:
- Consider adding more recent references, particularly from the last 5–10 years, to better position this study within the context of current research.
- Authors response: The manuscript includes a selection of the most relevant references, with 55% published within the past 10 years. In response to the reviewer’s suggestion, we incorporated additional recent publications to further support the study.
Introduction:
- It is unclear why only deoxynivalenol (DON) and nivalenol (NIV), among DON metabolites, were selected for analysis. Please clarify the rationale for focusing on these two.
- Author response: A description was added to the Materials and Methods section to emphasize this point.
- The novelty of the study should be more clearly articulated in relation to existing literature. How does this study differ from or advance previous findings?
- Author response: As presented in the seventh paragraph, the occurrence of Fusarium species is influenced by regional environmental factors. Considering that Rio Grande do Sul is the second-largest barley-producing state in Brazil, identifying the Fusarium species associated with Fusarium head blight (FHB) across its diverse physiographic regions is essential. This need is further underscored by the fact that previous studies have been conducted predominantly in the state of Paraná.
Methodology:
- Was a matrix effect evaluation in mycotoxin analysis conducted? If so, please include this information.
- During method validation for mycotoxin determination, both repeatability and recovery should be assessed and reported to ensure the accuracy and reliability of the results.
- Authors response: Dear reviewer, thank you for your comment. For improved understanding, we consider more appropriate to respond both comments in a common response. For this study, we did not validate a method for determination of mycotoxins in barley by HPLC-MS/MS. The barley samples were analyzed at the Laboratory of Mycotoxicological Analyses (LAMIC) at the Federal University of Santa Maria (UFSM) and the laboratory already has a validated methodology for the analysis of different mycotoxins in barley matrix by high performance liquid chromatography-tandem mass spectrometry (HPLC-MS/MS). The LAMIC quality system is designed to comply with CGCRE standards in accordance with NBR ISO/IEC 17025, which sets out the general requirements for the competence of testing and calibration laboratories. Thus, for the quantification of DON and NIV by HPLC-MS/MS in barley, the curve in the matrix is not prepared, but rather calibration curves using analytical standards and the linearity parameter is assessed and reported in a similar matrix (L. 231).
Results and Discussion:
- It would strengthen the discussion to compare the obtained mycotoxin concentrations (DON and NIV) with those reported in previous studies.
Authors: The discussion was improved by including comparisons with previously published concentration data from studies conducted in the Americas.
Reviewer 2 Report
Comments and Suggestions for Authors
The study by Furtado et al. offers a valuable contribution to the understanding of Fusarium head blight in barley grown under subtropical conditions. The integration of field sampling, molecular identification, and mycotoxin quantification is well executed, and the manuscript is clearly written and well organized. The following suggestions are provided with the aim of refining the focus and enriching the discussion.
-
The objectives of the study are well supported by the background presented in the introduction. However, it may help to state them more explicitly and concisely, possibly in a numbered format (e.g., i)… ii)… iii)…). This would clarify the scope and assist the reader in following the structure of the manuscript.
-
In the discussion, it could be helpful to include a short consideration of the toxigenic profiles of the Fusarium species identified. While F. graminearum is understandably central to the findings, species such as F. meridionale and F. poae are also isolated, yet their role is not addressed. A brief reference to their known capacity to produce DON or NIV in small grain cereals would improve the ecological context of the data.
-
The discussion might also benefit from a more general reflection on the variability of mycotoxin production across different species or isolates. Several studies have shown that this variability can be influenced by genetic factors, environmental conditions, and host-related parameters. Referring to these findings would help situate the results in a broader epidemiological and phytopathological framework (e.g. https://doi.org/10.1016/j.postharvbio.2023.112312; https://doi.org/10.1007/s42161-024-01751-8; https://doi.org/10.1016/j.micres.2021.126855)
The study is overall well conducted and informative. These are only suggestions aimed at enhancing clarity and scientific depth. It was a pleasure to review this manuscript!
Author Response
Dear Reviewer,
We sincerely thank you for your valuable comments and constructive suggestions regarding our manuscript.
The objectives of the study are well supported by the background presented in the introduction. However, it may help to state them more explicitly and concisely, possibly in a numbered format (e.g., i)… ii)… iii)…). This would clarify the scope and assist the reader in following the structure of the manuscript.
Authors response: Adjusted.
In the discussion, it could be helpful to include a short consideration of the toxigenic profiles of the Fusarium species identified. While F. graminearum is understandably central to the findings, species such as F. meridionale and F. poae are also isolated, yet their role is not addressed. A brief reference to their known capacity to produce DON or NIV in small grain cereals would improve the ecological context of the data.
Authors response: We did not assess the mycotoxin-producing capacity of the isolates, as this was beyond the scope of our study. We only referred to Fusarium graminearum due to the strong evidence observed in our results. In the final part of the discussion, we emphasized the need for further research on other mycotoxins and Fusarium species.
The discussion might also benefit from a more general reflection on the variability of mycotoxin production across different species or isolates. Several studies have shown that this variability can be influenced by genetic factors, environmental conditions, and host-related parameters. Referring to these findings would help situate the results in a broader epidemiological and phytopathological framework (e.g. https://doi.org/10.1016/j.postharvbio.2023.112312; https://doi.org/10.1007/s42161-024-01751-8; https://doi.org/10.1016/j.micres.2021.126855)
Authors response: As mentioned above, we do not have sufficient data to support a detailed discussion on the variability of mycotoxin production among different species or isolates. Experiments under controlled conditions may provide such information, and some of these studies are currently underway.
The study is overall well conducted and informative. These are only suggestions aimed at enhancing clarity and scientific depth. It was a pleasure to review this manuscript!
Authors response: Thank you for your suggestions!
Round 2
Reviewer 1 Report
Comments and Suggestions for Authors
The authors have taken the comments into account, and the manuscript is now suitable for acceptance.